# In vitro reconstitution of functional small ribosomal subunit assembly for comprehensive analysis of ribosomal elements in *E. coli*

Masaru Shimojo[1,2,6], Kazuaki Amikura [2,3,6], Keiko Masuda[1], Takashi Kanamori [4], Takuya Ueda[2,5] & Yoshihiro Shimizu [1✉]

In vitro reconstitution is a powerful tool for investigating ribosome functions and biogenesis, as well as discovering new ribosomal features. In this study, we integrated all of the processes required for *Escherichia coli* small ribosomal subunit assembly. In our method, termed fully Recombinant-based integrated Synthesis, Assembly, and Translation (R-iSAT), assembly and evaluation of the small ribosomal subunits are coupled with ribosomal RNA (rRNA) synthesis in a reconstituted cell-free protein synthesis system. By changing the components of R-iSAT, including recombinant ribosomal protein composition, we coupled ribosomal assembly with ribosomal protein synthesis, enabling functional synthesis of ribosomal proteins and subsequent subunit assembly. In addition, we assembled and evaluated subunits with mutations in both rRNA and ribosomal proteins. The study demonstrated that our scheme provides new ways to comprehensively analyze any elements of the small ribosomal subunit, with the goal of improving our understanding of ribosomal biogenesis, function, and engineering.

[1] Laboratory for Cell-Free Protein Synthesis, RIKEN Center for Biosystems Dynamics Research (BDR), Suita, Osaka 565-0874, Japan. [2] Department of Computational Biology and Medical Sciences, Graduate School of Frontier Sciences, The University of Tokyo, Kashiwa, Chiba 277-8562, Japan. [3] Department of Molecular Biophysics and Biochemistry, Yale University, New Haven, CT 06520, USA. [4] GeneFrontier Corporation, Kashiwa, Chiba 277-0005, Japan. [5] Department of Integrative Bioscience and Biomedical Engineering, Graduate School of Science and Engineering, Waseda University, Shinjuku, Tokyo 162-8480, Japan. [6] These authors contributed equally: Masaru Shimojo, Kazuaki Amikura. ✉email: yshimizu@riken.jp

Ribosomes are large macromolecular complexes that play central roles in cellular protein synthesis, the final step of gene expression. To elucidate their biogenesis and translational function, extensive in vitro reconstitution studies have been performed on the *Escherichia coli* ribosome, dating back to the early stages of the field of biochemistry[1,2]. These studies revealed the hierarchical assembly map of both 30S and 50S subunits[3,4]. Subsequently, these efforts were expanded to obtain more detailed knowledge regarding assembly pathways and kinetics, using cutting-edge technologies such as quantitative mass spectrometry and microscopic analyses[5,6]. Applied studies have also been performed to evolve ribosomes with improved functions or altered chemical properties[7,8]. Other efforts in the field of synthetic biology have been directed toward the design and construction of minimal cells, with the ultimate goal of answering questions such as "What is life?" and "What is the origin of life?"[9,10].

Currently, a variety of schemes are available for in vitro reconstitution of the *E. coli* ribosome. In particular, the integrated synthesis, assembly, and translation (iSAT) technology developed by Jewett et al. allows both 30S and 50S subunit assembly and detection of ribosomal activity in a sophisticated manner[11]. The method has made it possible to perform one-step rRNA transcription and ribosome assembly in a reaction mixture containing S150 crude cell extract under physiological conditions. The reconstitution efficiency of the iSAT method was subsequently improved by extensive optimization of the reaction conditions[12]. On the basis of this technology, a unique selection system using a liposome-sorting technique, which integrates iSAT with the protein synthesis using recombinant elements (PURE) reconstituted cell-free protein synthesis system[13], has been developed for the in vitro evolution of 16S rRNA[7].

The iSAT technology is based on a mixture of ribosomal proteins called TP30 (total proteins of the 30S subunit) or TP50 (total proteins of the 50S subunit). In contrast to this approach, in vitro ribosome reconstitution methods have also been developed using individually prepared ribosomal proteins, particularly for the 30S subunit. Using a conventional approach, including high salt concentration, heat activation, and ordered addition of ribosomal proteins according to the hierarchical assembly map revealed by Nomura and co-workers[3], Culver et al. demonstrated the in vitro reconstitution of the 30S subunit from individually purified recombinant ribosomal proteins and purified native 16S rRNA[14]. They also showed that the addition of protein chaperones facilitated assembly at low temperature[15]. Tamaru et al. adopted a similar approach using purified recombinant ribosomal proteins and showed that the addition of ribosome biogenesis factors such as Era and RsgA (YjeQ) facilitates reconstitution efficiency under low-salt conditions[16]. Li et al. succeeded in reconstituting the 30S subunit using purified native 16S rRNA with individually prepared ribosomal proteins in a cell-free manner utilizing the PURE system[17].

Based on these successful studies, it is clear that in vitro synthesized products can be used for both rRNA and ribosomal proteins, at least for the in vitro reconstitution of the 30S subunit. Hence, in this study, we integrate all of the previously developed approaches into a single reaction mixture based on the PURE system. Initially, we introduce the iSAT-like scheme into the previously developed system using purified recombinant ribosomal proteins[16], in which in vitro transcription of 16S rRNA, 30S subunit assembly from individually prepared ribosomal proteins and transcribed rRNA, and the subsequent superfolder GFP (sfGFP) synthesis reaction based on the assembled 30S subunits are coupled within the PURE system. The resultant system makes it possible to examine 16S rRNA mutant produced from DNA templates in the PURE system, similar to the approach described in previous studies[7,8]. With this system, we discover a partially orthogonal Shine–Dalgarno (SD)/anti-SD pair that can be used to differentiate "host" and "newly assembled" ribosomes in the PURE system. Using this new pair, we add the native 30S subunit to the system and replace each ribosomal protein with a DNA template encoding the corresponding ribosomal protein gene, and detect the activity of 30S subunits assembled from each cell-free-synthesized ribosomal protein based on the fluorescence of synthesized sfGFP. This system enable us to prepare and detect the activity of ribosomes with mutations in both 16S rRNA and ribosomal proteins. Moreover, this approach provides new ways to comprehensively analyze ribosomal elements in the 30S subunit, with the goal of elucidating ribosomal biogenesis, functions, and engineering.

## Results

**Fully recombinant-based small ribosomal subunits assembly.** We recently showed that active 30S subunits can be reconstituted from individually prepared recombinant ribosomal proteins under low-salt conditions using ribosome biogenesis factors[16]. In parallel studies, Jewett et al. developed the iSAT system, in which in vitro transcription of 16S rRNA, assembly of ribosomes from transcribed 16S rRNA and TP30, and protein synthesis from newly assembled ribosomes are performed in a single reaction mixture in the presence of S150 cell extract[11]. The presence of S150 cell extract was later shown to be nonessential because the iSAT system also functions in the PURE system[7]. These studies prompted us to develop an iSAT-like method based on individually prepared recombinant ribosomal proteins within the PURE system. We term this approach fully recombinant-based iSAT (R-iSAT).

R-iSAT was constructed in the PURE system using 50S subunits instead of 70S ribosomes, enabling simultaneous 16S rRNA transcription, ribosome assembly, and subsequent protein synthesis reaction using assembled 30S subunits (Fig. 1a). To detect the activity of the assembled ribosomes, we monitored changes in the fluorescence intensity of synthesize sfGFP over a time course. We compared native 16S rRNA, in vitro transcribed 16S rRNA prepared in advance, and co-transcriptionally synthesized 16S rRNA. In addition, we compared TP30 with the individually prepared recombinant ribosomal proteins.

A reconstitution experiment without 16S rRNA co-transcription revealed that either TP30 or recombinant ribosomal proteins, or either native 16S rRNA or in vitro transcribed 16S rRNA, could assemble into active 30S subunits capable of synthesizing sfGFP (Fig. 1b). The increase in fluorescence was initiated after a lag period that likely corresponds to the time required for ribosome assembly, sfGFP synthesis, and folding. This lag time was shortest when native 16S rRNA and TP30 were used, indicating that these reagents are optimal for the ribosome assembly reaction. Translation efficiency was also highest when native parts were used, probably due to the presence of modifications on both 16S rRNA and ribosomal proteins. Nonetheless, these results confirmed that active 30S subunits can be formed even with in vitro transcribed 16S rRNA and recombinant ribosomal proteins.

We then examined co-transcriptional assembly by replacing 16S rRNA with a DNA template encoding native rRNA derived from *rrnB* operon in *E. coli* genome (Fig. 1c). Under these conditions, sfGFP was synthesized both with TP30 and recombinant ribosomal proteins. The lag period was much longer than with native 16S rRNA, perhaps corresponding to the time required for rRNA transcription and subsequent assembly. We checked the final concentration of co-transcribed 16S rRNA by urea-polyacrylamide gel electrophoresis (PAGE) and found that

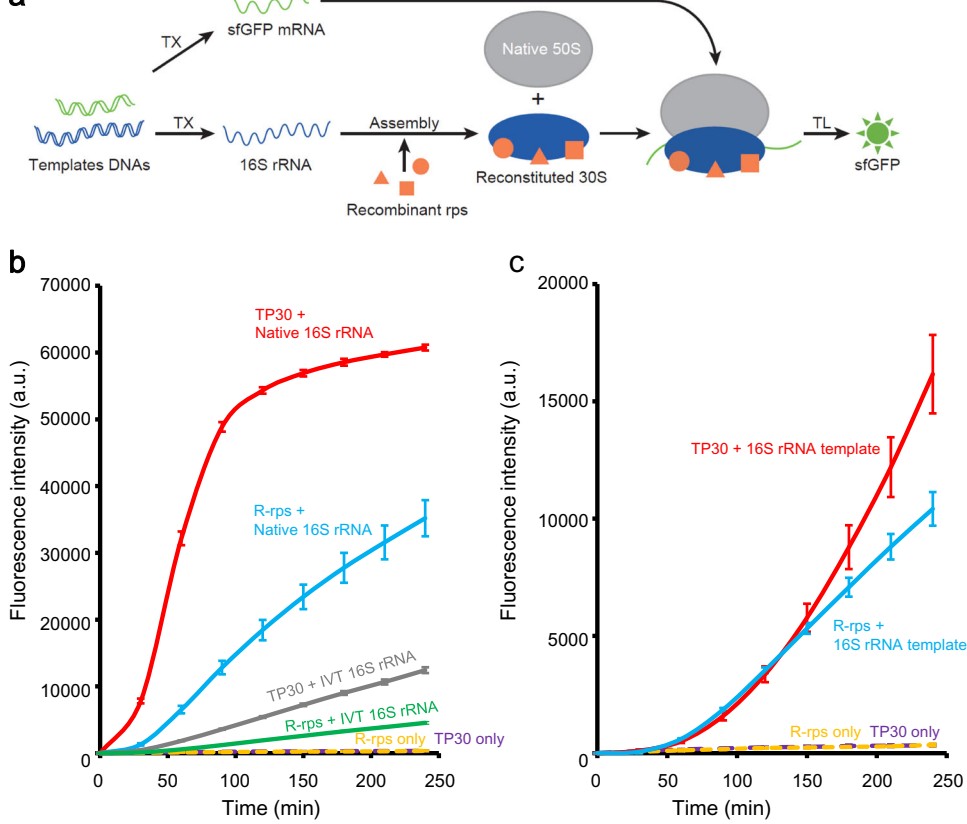

**Fig. 1 Fully Recombinant-based integrated synthesis, assembly, and translation (R-iSAT). a** Schematic of R-iSAT. Two template DNAs encoding 16S rRNA and sfGFP are added into the PURE system. Recombinant ribosomal proteins (recombinant rps) bind to the transcribed 16S rRNA, and they are assembled into 30S subunits that interact with native 50S subunits to form the ribosome. Transcribed sfGFP mRNA is translated on the ribosome to synthesize sfGFP synthesis. **b** Time-course analysis of sfGFP synthesis in reactions without coupling of 16S rRNA transcription. **c** Time-course analysis of sfGFP synthesis in reactions with coupling of 16S rRNA transcription. R-rps represents recombinant ribosomal proteins, and IVT 16S rRNA represents in vitro transcribed 16S rRNA. Fluorescence intensities after subtracting the background intensity are shown. Error bars indicate standard deviation of triplicate measurements.

sufficient rRNA (~0.8 μM), comparative to that in reaction without co-transcription (0.3 μM), was transcribed in the reaction mixture. Although the fluorescence intensity was lower than when native components were used, the results confirmed that the R-iSAT scheme is available for reconstitution of active 30S subunits. We note that the coupling of rRNA transcription with ribosome reconstitution was effective for increasing sfGFP synthesis when the final intensities are compared with each other (compare gray and green lines in Fig. 1b with red and blue lines in Fig. 1c, respectively). This may suggest that co-transcriptional folding of 16S rRNA in the presence of ribosomal proteins facilitated the assembly efficiency, as recently revealed[18,19]. Also note that a long-term analysis showed that R-iSAT continued for almost half a day with very slow kinetics (Supplementary Fig. 1). Concentrations of recombinant proteins were semi-optimized with a series of R-iSAT experiments (Supplementary Fig. 2a).

**Effects of ribosomal modification and biogenesis factors**. In a previous study, we performed matrix-assisted laser desorption ionization-time of flight mass spectrometry (MS) analysis of recombinant ribosomal proteins and showed that at least uS5 and bS18 are not acetylated and uS12 is not methylthiolated[16] (Here, we follow the new universal nomenclature for ribosomal proteins[20]). For deeper understanding of the modification status of recombinant ribosomal proteins, we performed native MS analysis of uS5, bS6, uS11, uS12, and bS18, which are known to be

posttranslationally modified[21] (Supplementary Fig. 3), demonstrating that large parts of these proteins are unmodified, consistent with previous results.

The previous study also showed that the addition of ribosome biogenesis factors facilitates 30S subunit reconstitution efficiency under low or moderate salt concentrations[16]. Hence, we reevaluated the effects of these factors on R-iSAT with semi-optimized concentration of the factors (Supplementary Fig. 2b). The results revealed trends similar to those observed in the previous study, in which Era had the greatest effect on total sfGFP synthesis (Supplementary Fig. 4). We also confirmed the overall effect of these factors in the R-iSAT without co-transcriptional coupling (Supplementary Fig. 5). Here, again note that the sfGFP synthesis efficiency was facilitated by the coupling of rRNA transcription. These data suggest that the effects of these factors were equivalent to those in our previous study, even though the method for detecting ribosome activity was switched from poly (U)-directed polyphenylalanine synthesis to sfGFP synthesis, and 16S rRNA transcription was coupled to the ribosome assembly reaction.

**Effects of removal of a single ribosomal protein**. One of the advantages of R-iSAT is that we can control the composition of ribosomal proteins in the reaction mixture. We examined the effects of removing individual ribosomal proteins from the R-iSAT to learn about each protein's role in assembly and activity.

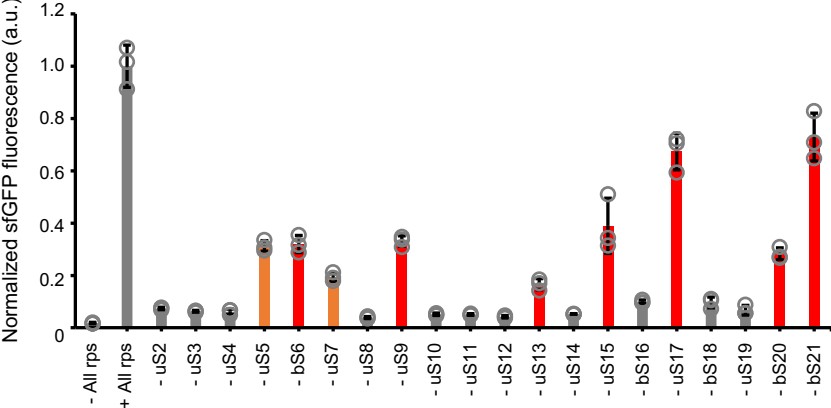

**Fig. 2 Effects of removal of a single ribosomal protein from R-iSAT.** Increase in sfGFP fluorescence after 4 h incubation was normalized by dividing by the average value of the control reaction. Red bars indicate nonessential ribosomal proteins, as revealed by genome deletion studies. Orange bars indicate putative nonessential ribosomal proteins identified in this study. Error bars indicate standard deviation of triplicate measurements. Each dot represents individual observed value.

**Table 1 Nonessential ribosomal protein genes, as determined by genome deletion studies.**

| Gene name | Protein | References |
|---|---|---|
| *rpsF* | bS6 | 22–24 |
| *rpsI* | uS9 | 22,24,25 |
| *rpsM* | uS13 | 22,24,27 |
| *rpsO* | uS15 | 24,28 |
| *rpsQ* | uS17 | 22,24,25 |
| *rpsT* | bS20 | 22–24 |
| *rpsU* | bS21 | 26 |

The results revealed that sfGFP was still synthesized in R-iSAT even in the absence of some proteins, including uS5, bS6, uS7, uS9, uS13, uS15, uS17, bS20, and bS21 (Fig. 2). Previous genomic deletion studies[22–28] showed that bS6, uS9, uS13, uS15, uS17, bS20, and bS21 are nonessential for viability (Table 1), consistent with our results in this study, although the extent to which the assembled subunits were functionally impaired differed among proteins. In addition, subunits without uS5 and uS7 were still active for sfGFP synthesis in R-iSAT. Liquid chromatography–MS (LC–MS) analyses revealed no carry-over of ribosomal proteins from any other components of the R-iSAT mixtures (Supplementary Data 1), suggesting that these results reflected the nonessentiality of these proteins for ribosomal assembly and function. Genomic deletion approaches aimed at elucidating the function of each ribosomal protein may be limited by the fact that their outcomes can depend on the relationship between growth rate and ribosomal activities or gene regulation. These findings demonstrate that R-iSAT makes possible more detailed analyses of each ribosomal protein. The dependence of bS1 was also examined and found that reconstituted bS1-free ribosomes synthesized sufficient amount of sfGFP in R-iSAT (Supplementary Fig. 6a). This appears to be inconsistent with our previous results where bS1-free ribosomes were not functional for the protein expression from mRNA with 5′ leader sequence[29]. This will be mentioned later in this paper.

**Differentiation of "host" and "newly assembled" ribosomes.** In this study, we sought to couple ribosomal protein synthesis in R-iSAT. Hence, we searched for a suitable orthogonal SD/anti-SD pair that would enable us to differentiate "host" from "newly assembled" ribosomes using the R-iSAT feature of 16S rRNA co-transcriptional synthesis. To this end, we first selected pairs from a previous network study of orthogonal ribosome/mRNA pairs based on in vivo selection experiments[30]. In particular, we selected 16S rRNA sequences for rRNA-2, rRNA-8, and rRNA-9 and SD sequences for mRNA-A, mRNA-B, and mRNA-C, assuming that each of the respective pairs is orthogonal (rRNA-2/mRNA-A, rRNA-8/mRNA-B, and rRNA-9/mRNA-C, respectively) (Fig. 3a). In addition to the anti-SD region, mutations were introduced in the 16S rRNA sequences at positions 722 and 723, which form a bulge proximal to the minor grove of the SD/anti-SD helix.

All possible combinations, including wild-type pairs, were tested in R-iSAT, in which sfGFP synthesis was regulated by the selected SD sequences. Although the results suggested that it was difficult to achieve complete orthogonality (Fig. 3b, c), partial orthogonality was apparent: sfGFP synthesis under non-wild-type SD sequences was less active when the wild-type 30S subunits were used (see green box in Fig. 3b). Nevertheless, sfGFP synthesis was still observed in the presence of selected non-wild-type SD sequences (9.1–11.4% relative to wild-type SD). Because we wanted to minimize background sfGFP synthesis with wild-type ribosomes as depicted in Fig. 4a, we additionally designed an SD sequence (mRNA-comp) complementary to the strong SD sequence frequently used in overexpression vectors such as pET (Merck Millipore). The anti-SD sequence (rRNA-comp) was also designed to be complementary to the wild type (Fig. 3a). Surprisingly, sfGFP synthesis from mRNA-comp by the wild-type 30S subunit was suppressed to the background level (2.4%; green box in Fig. 3b), whereas the 30S subunit with rRNA-comp tended to accept mRNAs with a variety of SD sequences (blue box in Fig. 3b). These results demonstrated that sfGFP synthesis from mRNA-comp was stimulated only by the 30S subunit with rRNA-comp (see red box in Fig. 3c). The ratio of sfGFP expression against the expression with the wild-type ribosome was highest among tested orthogonal pairs (Supplementary Fig. 7). The observed one-sided orthogonality (blue box in Fig. 3b vs. red box in Fig. 3c) provided the R-iSAT with an ideal feature for differentiating "host" from "newly assembled" ribosomes using this novel pair.

It is interesting to note that unexpected results were obtained through this series of experiments. The levels of sfGFP synthesis using non-wild-type pairs were higher than the wild-type pair.

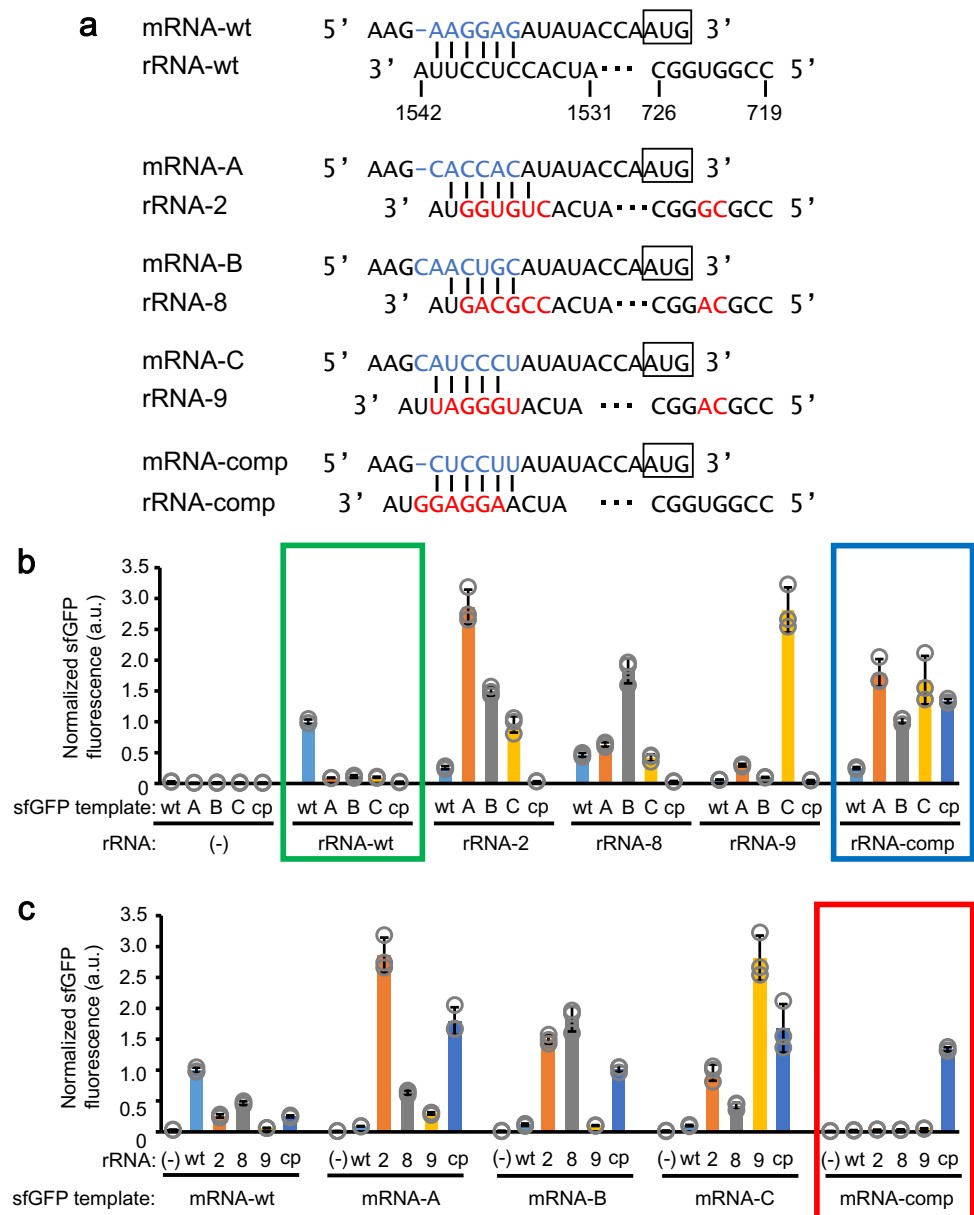

**Fig. 3 A mutational study of 16S rRNA to search for applicable orthogonal SD/anti-SD pair for differentiating "host" and "newly assembled" ribosomes.** **a** Sequences of 16S rRNA and mRNA examined in this study. SD sequences are indicated by light blue characters, and mutated nucleotides in 16S rRNA by red characters. **b**, **c** Synthesis of sfGFP using various SD/anti-SD pairs. Increase in sfGFP fluorescence after 4 h incubation was normalized by dividing by the average value of the control reaction. Graphical display is different, but the same data were used in (**b**) and (**c**). Symbols "wt", "A", "B", "C", and "cp" for sfGFP template represent mRNA-wt, mRNA-A, mRNA-B, mRNA-C, and mRNA-comp, respectively, as shown in (**a**). Symbols "wt", "2", "8", "9", and "cp" for rRNA represent rRNA-wt, rRNA-2, rRNA-8, rRNA-9, and rRNA-cp, respectively, as shown in (**a**). A green box indicates a data set obtained using rRNA-wt, a blue box indicates a data set obtained using rRNA-comp, and a red box indicates a data set obtained using mRNA-comp. Error bars indicate standard deviation of triplicate measurements. Each dot represents individual observed value.

Particularly, mRNA-A/rRNA-2 and mRNA-C/rRNA-9 showed threefold higher sfGFP fluorescence compared to the wild-type pair. Time-course of the sfGFP synthesis clearly showed that the use of non-wild-type pairs are beneficial for increasing sfGFP synthesis in R-iSAT (Supplementary Fig. 8), suggesting that mutations in the anti-SD sequence, together with those at positions 722 and 723, facilitated the ribosome assembly or the gene expression processes. In particular, three pairs from the literature[30] were selected through in vivo evolutionary experiments where a plenty of wild-type ribosomes coexisted in cells. This might have resulted in the selection of sequences that are favorable for the assembly or

translation efficiency. Because the mutation in the anti-SD sequence may affect all gene expression in cells, this region may have been left out during the evolution, leaving room for improving ribosome functions. This point is insightful for the studies for ribosome engineering and ribosome evolution, which should be investigated more in detail in the future.

**Co-translational coupling of ribosomal protein synthesis.** Finally, we sought to integrate ribosomal proteins synthesis in R-iSAT. A template DNA encoding specific ribosomal proteins under the control of the wild-type SD sequence were added to the

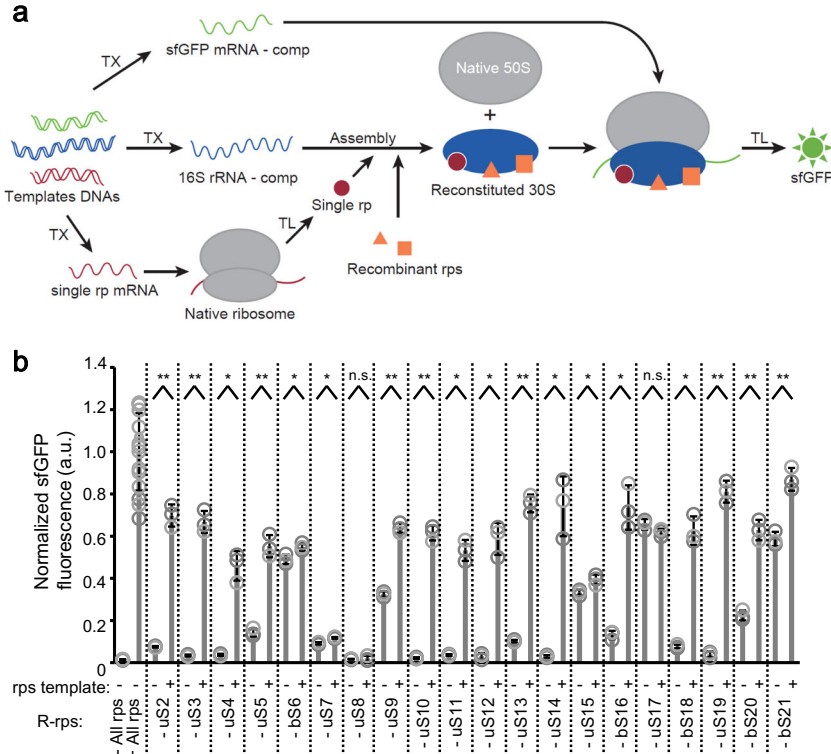

**Fig. 4 Co-translational coupling of ribosomal proteins synthesis in R-iSAT. a** Schematic of R-iSAT coupled with ribosomal proteins synthesis. Three template DNAs encoding 16S rRNA-comp (shown in Fig. 3a), a ribosomal protein under the wild-type SD, and sfGFP under the SD encoded in mRNA-comp (shown in Fig. 3a) were added into the PURE system. A ribosomal protein synthesized by the native ribosome and recombinant ribosomal proteins (Recombinant rps) in the reaction mixture bind to transcribed 16S rRNA-comp, and these components are assembled into 30S subunits that interact with native 50S subunits to form a partially orthogonal ribosome for subsequent sfGFP synthesis. Transcribed sfGFP mRNA-comp is translated on the newly assembled ribosome to synthesize sfGFP. **b** Effects of ribosomal protein synthesis in R-iSAT. The increase in sfGFP fluorescence after 4 h incubation was normalized by dividing by the average value of the control reaction. Error bars indicate standard deviation of at least triplicate measurements. Each dot represents individual observed value. Double asterisk, single asterisk, and n.s. indicate that $P$ values are less than 0.003, $P$ values are less than 0.05, and $P$ values are more than 0.05, respectively. Welch's $t$ test was applied between in the presence and absence of the rps template.

R-iSAT mixture in place of specific ribosomal proteins, and were synthesized by the native 30S subunits. Synthesized ribosomal proteins were designed to be assembled with rRNA-comp, followed by evaluation of sfGFP synthesis from mRNA-comp by the newly assembled 30S subunits (Fig. 4a).

We observed an increase in sfGFP synthesis when the putative essential proteins identified in Fig. 2 were synthesized (uS2, uS3, uS4, uS10, uS11, uS12, uS14, bS16, bS18, and uS19), with the exception of uS8 (Fig. 4b). Synthesis of sfGFP was observed in the absence of nonessential proteins (bS6, uS9, uS13, uS15, uS17, bS20, and bS21) and the putative nonessential proteins identified in Fig. 2 (uS5 and uS7). A prominent increase in ribosomal protein synthesis was observed for uS5, uS9, uS13, bS20, and bS21, whereas only a slight or no increase was observed for bS6, uS7, uS15, and uS17 (Fig. 4b). The expression of each ribosomal protein in the PURE system was confirmed ahead and the data showed successful expression of all proteins with yields in the same orders of magnitude without any aggregate formations (Supplementary Fig. 9).

The cell-free-synthesized essential protein uS8 and nonessential proteins bS6, uS7, uS15, and uS17 appeared to be nonfunctional in the assay. Although it is possible that we could not detect the activity of these proteins due to problems such as misfolding or low yield, it is also possible that the timing of their synthesis affects the assembly process. Interestingly, four of these five proteins (uS7, uS8, uS15, and uS17), which showed slight or no increase in sfGFP fluorescence are identified as primary

binding proteins in Nomura's assembly map (Supplementary Fig. 10); the fifth protein, bS6, is considered to be a secondary binding protein dependent on uS15. Because production of sufficient protein for 30S subunit assembly will inevitably be delayed if the proteins must be simultaneously synthesized, it is possible that partially assembled subunits are caught in kinetic traps that can no longer bind these proteins.

To further clarify this point, we performed two-step reactions in which ribosomal proteins were translated in the first step (2 h) in the presence of recombinant ribosomal proteins, and then rRNA transcription and subsequent sfGFP synthesis occurred in the second step (4 h). The results revealed that pre-synthesis of uS7, uS8, and uS15 facilitated active 30S subunit assembly, whereas bS6 and uS17 yielded results similar to those of simultaneous synthesis (Supplementary Fig. 11), confirming the functionality of cell-free-synthesized uS7, uS8, and uS15. uS8 was particularly important: almost no functional subunits were produced when the supply of uS8 was delayed. We further tested uS8 by performing two-step reactions in which 16S rRNA co-transcription was performed in the first step, and recombinant ribosomal proteins were added in the second step. Again, almost no functional subunits were produced, suggesting that the timing of uS8 binding to the 16S rRNA plays a crucial role as a checkpoint that determines the fate of the assembled subunits (Supplementary Fig. 12).

We also examined bS1 expression in this system but effect was comparatively lower than other proteins (Supplementary Fig. 6b).

This might be caused by the decrease in rRNA or sfGFP mRNA synthesis due to the competition among RNAs to be transcribed. This was suggested by an experiment with sufficient amount of bS1, which resulted in decrease of sfGFP synthesis when bS1 template is added (Supplementary Fig. 6b). Because the size of bS1 is much larger than other proteins, degree of this competition might become higher and thus, the effect of bS1 expression might have been constrained. Nevertheless, we observed slight stimulation of sfGFP synthesis when the bS1 template was added, suggesting functional bS1 expression.

It was surprising that sfGFP synthesis was highly dependent on the presence of bS1 in this experiment, which was inconsistent with the result using wild-type rRNA (Supplementary Fig. 6a). Because the different point was only the use of the orthogonal pair (rRNA-comp/mRNA-comp) for sfGFP expression, bS1 dependency might be changed according to the selected SD/anti-SD pair. We also note that we have previously showed that bS1-free ribosomes were not functional for protein expression from mRNA with 5′ leader sequence[29]. The leader sequence including SD sequence was again different from the one used in this ppaer, suggesting that the bS1 dependency depends on the selected leader sequence. Detailed analysis should be performed for clarifying this point in the future.

**Mutational analysis of the ribosomal protein**. Taking advantage of the co-translational synthesis of ribosomal proteins in R-iSAT, we performed a mutational study of uS12, which along with critical nucleotides of 16S rRNA plays an important role in the decoding process[31]. Mutations in uS12 affect translational accuracy, and are associated with resistance or dependence on the antibiotic streptomycin, which promotes misreading of the genetic code by the ribosome[32].

Several uS12 mutants, including K42T, P90L, and G91D, were synthesized in R-iSAT without uS12. The K42T mutant is streptomycin-resistant, whereas P90L and G91D are streptomycin-dependent[33]. To detect streptomycin dependence, we performed two-step reactions: uS12 synthesis and its assembly into the 30S subunit with rRNA-comp occurred in the first step (2 h); next, to allow sfGFP synthesis by the assembled 30S subunits, streptomycin and sfGFP template DNA were added in the second step (4 h).

When wild-type uS12 was synthesized, sfGFP fluorescence was dramatically reduced upon addition of streptomycin (Fig. 5). By contrast, sfGFP fluorescence was not affected by addition of streptomycin when uS12 mutants were synthesized, consistent with the previously reported streptomycin-resistant properties of these mutants[33]. Surprisingly, P90L and G91D exhibited no streptomycin dependence: the same level of sfGFP synthesis was observed even in the absence of streptomycin. This is consistent with a previous report that sufficient amounts of proteins can be synthesized by these mutants in a cell-free manner in the absence of streptomycin[33]. Thus, the reported drug-dependent properties of strains carrying these mutations may be due to factors other than ribosomal activity. The results shown here demonstrated that a variety of mutants in ribosomal proteins can be easily tested using R-iSAT coupled with co-translational ribosomal protein synthesis; by contrast, conventional methods of testing require multiple processes, including laborious genetic manipulation.

## Discussion

Multiple approaches for reconstitution of both ribosomal subunits have been proposed to date. Post-transcriptional modifications on a specific region in 23S rRNA was suggested to be required for the functional 50S subunit assembly using in vitro transcribed rRNAs[34,35], whereas this drawback was overcome by the construction of iSAT supplemented with S150 cell extract, which may include enzymes for those modifications[11]. Contrastingly, functional 30S subunits can be assembled with in vitro transcribed rRNA without any modifications[7,36]. Consistent with this, in the present study, we demonstrated that functional 30S subunits capable of synthesizing sfGFP can be assembled by using in vitro synthesized 16S rRNA and native ribosomal protein mixtures, so-called TP30 (Fig. 1b). This suggests that 16S rRNA modifications are nonessential for the 30S subunit's function. The notice for importance of modifications are expanded to ribosomal proteins. We successfully replaced TP30 with a set of individually prepared recombinant ribosomal proteins (Fig. 1). Native MS analyses showed that large parts of prepared proteins are unmodified (Supplementary Fig. 3), implying that functional 30S subunits may not require any modifications for the protein synthesis as discussed previously[21].

In addition to enjoying iSAT's flexibility with regard to rRNA synthesis[11], R-iSAT neither use cell extract nor TP30, and

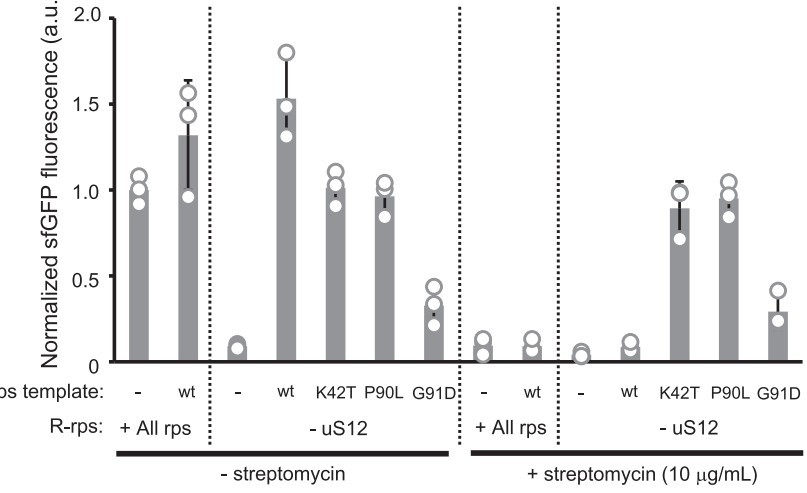

**Fig. 5 Mutational analysis of uS12.** Reactions were separated into two steps: uS12 synthesis and the 30S subunit assembly were performed in the first step (2 h), and then streptomycin and sfGFP template DNA were added for sfGFP synthesis in the second step (4 h). The increase in sfGFP fluorescence after the 4-h incubation in the second step was normalized by dividing by the average value of the control reaction. Error bars indicate standard deviation of at least triplicate measurements. Each dot represents individual observed value.

consequently is more flexible with respect to ribosomal protein composition, as well as testing of ribosome biogenesis factors and modification enzymes. The sfGFP synthesis rate in R-iSAT was lower than in reactions using native components, presumably due to the lack of modifications in both 16S rRNA and ribosomal proteins (Fig. 1). Ribosome biogenesis factors facilitate 30SS subunit assembly in R-iSAT (Supplementary Fig. 4), consistent with our previous observations using native 16S rRNA[16]. In addition to these factors, we can test 16SS rRNA modification enzymes with their substrates; this should be evaluated in future work to construct an even more efficient ribosome assembly method.

Removal of single ribosomal proteins from the R-iSAT showed that sfGFP synthesis could proceed even in the absence of uS5, bS6, uS7, uS9, uS13, uS15, uS17, bS20, and bS21 (Fig. 2). Except for uS5 and uS7, all of those proteins have already been shown by genomic deletion studies to be nonessential for ribosomal function (Table 1); however, the nonessentiality of uS5 and uS7 is an interesting finding. Mutations that affect important interactions between uS4 and uS5 are closely related to the ribosome ambiguity or *ram* phenotype, which increases the error rate of translation[37]. uS7 interacts with uS11 and E-site tRNA[37] and a mutational study of uS7 revealed that appropriate contact between uS7 and uS11 is important for translational fidelity[38]. Therefore, these proteins are likely to be involved in translational fidelity, and it would be interesting to measure the fidelity of ribosomes lacking uS5 or uS7 assembled by R-iSAT. We note that similar studies were of course performed by Nomura and co-workers using poly(U)-directed polyphenylalanine synthesis[39,40] and there are some discrepancies between their results and ours. For example, the subunit without bS16 or bS18 showed relatively high activity in their analysis whereas lack of these proteins showed almost background level activity in the present study. Although such discrepancy may arise from the difference of mRNA (poly(U) vs. sfGFP mRNA) used, leaky sfGFP synthesis was also observed in bS16- and bS18-lacking R-iSAT (Fig. 2 and Fig. 4b) and hence, it does not disagree with their suggestion that these proteins are not directly involved in ribosomal function but important for ribosomal assembly and structural stabilization.

As noted by Nomura, the removal of single ribosomal proteins from R-iSAT may also affect the ribosome assembly process, rather than the functionality of the assembled ribosome, i.e., the composition of the assembled ribosomes may be dramatically altered by removal of a single protein. To elucidate whether such removals affect assembly or translation itself, it will be necessary to quantify ribosomal protein composition using techniques such as quantitative MS and Cryo-EM. We are currently undertaking these studies using a protein quantification method we developed previously[41].

Our results also suggest that assembly can proceed via multiple pathways, as proposed previously[42,43]. Nonessential proteins uS15, uS17, bS20, and the putative nonessential protein uS7 are classified as primary binding proteins in the assembly map (Supplementary Fig. 10), and may thus affect subsequent protein binding. Nevertheless, functional subunit assembly occurs even in their absence, suggesting the presence of alternative routes for formation of functional subunits. Thus, we may be able to control the route to mature subunits via parallel pathways by omitting specific ribosomal proteins. The testing of ribosome biogenesis factors for maturation and modifications in such controlled assembly reactions represents an important aspect of this study, and may provide new insights into ribosomal function and assembly processes.

Further integration of the co-translational synthesis of ribosomal proteins with R-iSAT, applying the one-sided orthogonal pair discovered by both the SD and 16S rRNA mutational studies (Fig. 3), will provide ways to investigate ribosomal protein

function (Fig. 4). We observed a dramatic increase in sfGFP synthesis in the context of coupled synthesis ribosomal proteins, with the exception of bS6, uS7, uS8, uS15, and uS17. Further investigation using two-step reactions revealed that pre-synthesis of uS7, uS8, and uS15 is necessary for activity (Supplementary Fig. 11). All of these proteins are classified as primary binding proteins in the assembly map (Supplementary Fig. 10), and our data suggest that the timing of synthesis of these proteins is important for their participation in the assembly process, particularly for uS8 (Supplementary Figs. 11 and 12). The importance of uS8 for the assembly process was also suggested by feedback regulatory mechanism in *spc* operon[44], where uS8 binds directly to the mRNA to control the expression of operon genes. Eleven from twelve genes are ribosomal proteins and thus, uS8 plays a role in balancing the ribosomal proteins expression in cells. On the other hand, for other proteins such as uS4, uS9, uS13, bS16, bS18, uS19, and bS20, a delay in synthesis did not affect functional subunit assembly, irrespective of whether they were primary or secondary binding proteins (Supplementary Fig. 10). It remains unclear whether these proteins participate in assembly in a different order than in the assembly map, or if the assembling particles can maintain their states while waiting for these proteins to bind. However, it is likely that the assembly process is robust enough to tolerate a delay in synthesis of these proteins.

The scheme presented in this study provides a platform for mutational studies aimed at comprehensive analysis of 30S subunit elements, including the 16S rRNA and all ribosomal proteins. In addition to the mutational studies on 16S rRNA performed with the iSAT[7,8], R-iSAT enables studies of both 16S rRNA (Fig. 3) and ribosomal proteins (Fig. 5) by using DNAs encoding the mutations of interest. Although the functions of ribosomal proteins such as uS4, uS5, and uS12, which are located around the decoding center, are well known, other proteins may also affect subunit assembly and subsequent translational processes. Furthermore, it may become possible to simultaneously evolve both 16S rRNA and ribosomal proteins by introducing mutations into both DNA templates. Such ribosome engineering studies could contribute to the construction of more efficient ribosome assembly systems in the future.

## Methods

**Native ribosomal components**. 70S ribosomes were prepared from *E. coli* A19 cells by hydrophobic chromatography followed by ultracentrifugation as described previously[45]. They were further purified by sucrose density gradient (SDG) centrifugation as previously described[46]. Ribosomes pelleted by ultracentrifugation were dissolved in ribosome buffer (10 mM HEPES-KOH, pH 7.6, 30 mM KOAc, 10 mM Mg(OAc)$_2$, and 1 mM DTT) and stored at −80 °C. Ribosomal subunits were prepared as follows: 70S ribosomes were dissociated by dialysis against low ribosome buffer with low Mg$^{2+}$ concentration (1 mM Mg(OAc)$_2$) using Slide-A-Lyzer Dialysis Cassette or MINI Dialysis Units (3500 MWCO; Thermo Fisher Scientific, USA). The resultant mixtures were subjected to 12.2–36.6% SDG centrifugation in SDG buffer (10 mM HEPES-KOH, pH 7.6, 1 mM Mg(OAc)$_2$, 60 mM NH$_4$Cl, and 2 mM DTT), and the fractions corresponding to the 30S and 50S subunits were collected and pelleted by ultracentrifugation. The subunits were dissolved in ribosome buffer and stored at −80 °C. SDG separation had to be performed three times to obtain 50S subunits without any carry-over of 30S subunits. Native 16S rRNA was extracted from 30S subunits using phenol and chloroform, followed by isopropanol precipitation. Precipitated rRNA was purified using Micro Bio-Spin columns (Bio-Rad, USA), using water for elution, and stored at −80 °C. TP30 was extracted from 30S subunits as previously described[46] and stored at −80 °C.

**Recombinant ribosomal proteins and biogenesis factors**. Purification procedures for ribosomal proteins and ribosome biogenesis factors were described previously[16]. Before the R-iSAT reactions, bS1 was dialyzed against bS1 buffer (50 mM HEPES-KOH, pH 7.6, 100 mM K-Glu, 10 mM Mg(OAc)$_2$, 30% glycerol, and 7 mM 2-mercaptoethanol) and equimolar amounts of ribosomal proteins uS2–bS21 were mixed and dialyzed against R-rps buffer (50 mM HEPES-KOH, pH 7.6, 300 mM K-Glu, 30% Glycerol, and 2 mM DTT) using Slide-A-Lyzer MINI Dialysis Units (3,500 MWCO). Ribosome biogenesis factors including Era, RbfA,

RimI, RimJ, RimM, RimN, RimP, and RsgA (YjeQ) were dialyzed against rbf buffer (HEPES-KOH, pH 7.6, 100 mM KOAc, 30% Glycerol, and 1 mM DTT).

**DNA templates**. DNA sequences of the plasmids used in preparation of DNA templates for R-iSAT are summarized in Supplementary Data 2. The 16S rRNA gene was amplified by polymerase chain reaction (PCR) from the *rrnB* operon in *E. coli* and cloned into pET15b (Merck Millipore, USA) directly downstream of the T7 promoter using the Gibson Assembly system (New England Biolabs, USA). The sequences encoding sfGFP and each ribosomal protein were cloned into pET32b and pET26b (Merck Millipore), respectively. Only a gene for ribosomal protein bS1 was cloned into pQE30 (QIAGEN, Germany) Genes encoding 16S rRNA mutants reported by Rackham and Chin[30] and uS12 mutants reported by Chumpolkulwong et al.[33] were generated by PCR-based site-directed mutagenesis method using the PrimeSTAR Max DNA Polymerase and *Dpn*I (Takara, Japan). DNA templates for 16S rRNA transcription and sfGFP and ribosomal protein translation in R-iSAT were prepared by 1- or 2-step PCR from plasmids encoding these genes using adequate primers listed in Supplementary Data 3. Replacement of the SD sequence for sfGFP expression was carried out by the PCR. A template for 16S rRNA-comp with a mutation only in anti-SD was generated by replacing the reverse primer with one encoding a sequence complementary to mutated anti-SD. Amplified DNA templates were purified using Wizard SV Gel and PCR Clean-Up System (Promega, USA) and eluted with water. We note that we can share all of the plasmids on requests.

**Ribosome assembly by iSAT or R-iSAT**. Reaction mixtures (10 μL) contained solutions I (Buffer Mix) and II (Enzyme Mix) of PUREfrex 2.0 (GeneFrontier Corporation); 0.3 μM 50S subunits; 20 nM DNA template encoding 16S rRNA, 0.3 μM native 16S rRNA, or 0.3 μM in vitro transcribed 16S rRNA; 5 nM DNA template encoding sfGFP; 0.5 μM recombinant bS1 protein; 0.5 μM each of recombinant ribosomal proteins uS2–bS21; and 0 or 0.125 μM of each biogenesis factor. For negative controls, an equal volume of the same buffer was added to reaction mixtures to rule out the influence of buffers. Reactions were carried out at 37 °C for 4 h in Mx3005P (Agilent Technologies, USA), and sfGFP fluorescence was monitored during incubation.

**Native MS analysis**. Native MS analysis was performed using an Orbitrap mass spectrometer (Q Exactive; Thermo Fisher Scientific) equipped with a nanospray ion source (Nanospray Flex, Thermo Fisher Scientific). Ribosomal proteins (uS5, bS6, uS11, uS12, and bS18) at a concentration of 1 mg/mL were dialyzed against water and they were mixed with an equal volume of a solution containing 50% acetonitrile and 1% acetic acid. One microlitre of each sample was filled into the Cellomics tips (CT-10 μm, HUMANIX, Japan) and electrosprayed at spray voltage 1.7 kV into the MS (positive mode, scan range of 400–2000 m/z, 140,000 FWHM resolution). Theoretical spectra were obtained using Xcalibur (Thermo Fisher Scientific).

**LC–MS analysis**. Preparation of samples for mass spectrometric analysis was performed according to a phase transfer surfactant (PTS)-aided protocol[47]. Ribosomal protein mixtures, including 5 pmol each of recombinant ribosomal proteins (uS2–bS21) or no specified ribosomal protein, 5 pmol of bS1, 3 pmol of 50S subunits, and solution II of PUREfrex 2.0 (GeneFrontier Corporation) were dissolved in 10 μL of a PTS buffer (10 mM sodium deoxycholate, 10 mM sodium N-lauroylsarcosinate, and 50 mM NH₄HCO₃). They were reduced with 10 mM TCEP at 37 °C for 30 min, alkylated with 20 mM iodoacetamide at 37 °C for 30 min, and quenched with 20 mM L-cysteine residues. Before digestion, the sample was diluted 5-fold with 50 mM NH₄HCO₃ and digestion was performed by incubation with 100 ng of Lys-C (FUJIFILM Wako Chemicals, Japan) and 100 ng of trypsin (Thermo Fisher Scientific) at 37 °C overnight. After digestion, 5 μL of 10% trifluoroacetic acid (TFA) was added to the samples to precipitate the detergents. Supernatants of centrifuged samples were desalted using self-prepared stage tips[48] and dried on a SpeedVac. Mass spectrometric analysis was performed using an Orbitrap mass spectrometer (LTQ Orbitrap Velos Pro, Thermo Fisher Scientific) equipped with a nanospray ion source (Nanospray Flex, Thermo Fisher Scientific) and a nano-LC system (UltiMate 3000, Thermo Fisher Scientific). The dried peptide mixtures were dissolved in 10 μL of a solution containing 5% acetonitrile and 0.1% TFA, and 1 μL of each sample was applied to the nano-LC system. Peptides were concentrated using a trap column (0.075 × 20 mm, 3 μm, Acclaim PepMap 100 C18, Thermo Fisher Scientific) and then separated using a nano capillary column (0.1 × 150 mm, 3 μm, C18, Nikkyo Technos) using two mobile phases A (0.1% formic acid) and B (acetonitrile and 0.1% formic acid) with a gradient (5% B for 5 min, 5–45% B in 40 min, 45–90% B in 1 min, and 90% B in 4 min) at a flow rate of 500 nL/min. Elution was directly electrosprayed (2.2 kV) into the MS (positive mode, scan range of 200–1500 m/z, 60,000 FWHM resolution at 400 m/z). Data analysis was performed using Proteome Discoverer 2.2 (Thermo Fisher Scientific) to select peptides specific for each ribosomal protein. Peptides with sufficient peptide spectrum matches (two to five peptides for each protein) were selected and quantified using Skyline[49] (v4.2.0.18305; MacCoss Lab Software). Peak areas were calculated by setting the MS1 filter to a count of three (M, M + 1, and M + 2).

**R-iSAT coupled with ribosomal protein synthesis**. Reaction mixtures (10 μL) contained solutions I and II of PUREfrex 2.0 (GeneFrontier Corporation), 0.03 μM 30S subunit, 0.3 μM 50S subunit, 20 nM DNA template encoding

rRNA-comp, 2 nM DNA template encoding sfGFP-comp, 4 nM DNA template encoding the ribosomal protein to be synthesized, 0.5 μM recombinant S1 protein, and 0.5 μM each of recombinant ribosomal proteins uS2–bS21 without the protein that was to be removed. Reactions were carried out at 37 °C for 4 h in Mx3005P (Agilent Technologies), and sfGFP fluorescence was monitored during incubation.

**SDS-PAGE analysis of PURE synthesized ribosomal proteins**. Each ribosomal protein was expressed in PUREfrex 2.0 (GeneFrontier Corporation). Reaction mixtures (20 μL) contained solutions I (Buffer Mix) and II (Enzyme Mix) from PUREfrex 2.0, 2 μM 70SS ribosomes, 1 nM DNA templates encoding ribosomal proteins, and 0.2 μM [³⁵S] methionine (PerkinElmer, USA). After incubation at 37 °C for 2 h, aliquots (7 μL) were withdrawn as total reaction mixtures (T) and remaining solutions were centrifuged at 21,600g for 30 min and then supernatant fractions (S) were withdrawn. Totally, 1.33 μL of each aliquot was analyzed with 19% SDS-PAGE and the gel image was visualized by a BAS-5000 bio-imaging analyzer (GE Healthcare, USA).

**Mutational analysis of uS12 with the R-iSAT**. Reaction conditions were based on R-iSAT with ribosomal protein synthesis, but two-step reactions were performed for the mutational study. Translation of uS12 and subsequent assembly into 30S subunits were performed at 37 °C for 2 h in the first step, and then DNA templates for sfGFP-comp were added with or without 10 μg/mL streptomycin sulfate (FUJIFILM Wako Chemicals) for the second-step reactions. For negative controls, an equal volume of water was added to reactions. Second-step reactions were carried out at 37 °C for 4 h in Mx3005P (Agilent Technologies), and sfGFP fluorescence was monitored during incubation.

**Statistics and reproducibility**. Error bars indicate standard deviation of at least triplicate measurements. Welch's *t* test was applied to show statistical significance using values of at least triplicate measurements. Exact number of replicates are shown in Supplementary Data 4.

**Reporting summary**. Further information on research design is available in the Nature Research Reporting Summary linked to this article.

## Data availability

Data for mass spectrometric analysis of protein mixtures used in R-iSAT is available in Supplementary Data 1. DNA sequences of the plasmids and DNA primers used in preparation of DNA templates for R-iSAT is available in Supplementary Datas 2 and 3, respectively. Source data for main figures are presented in Supplementary Data 4.

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

## Acknowledgements

The authors thank Nono Takeuchi-Tomita for helpful discussions. This work was supported by a scholarship from The Futaba Foundation (to M.S.), a Grant-in-Aid in number 17H05680 (to Y.S.) and 15K16083 (to K.A.) from Japan Society for the Promotion of Science (JSPS), Human Frontier Science Program (RGP0043/2017 to Y.S.), Astrobiology Center Project of the National Institutes of Natural Sciences (AB271004 and AB281007 to K.A. and AB311005 to Y.S. and K.A.), and an intramural Grant-in-Aid from the RIKEN Center for Biosystems Dynamics Research (to Y.S.).

## Author contributions

M.S., K.A., T.U., and Y.S. designed the study. M.S., K.A., and Y.S. wrote the paper. M.S. performed most of the experiments and K.A. prepared the recombinant ribosomal proteins. M.S., K.A., and Y.S. interpreted the data. M.S. and K.M. performed the mass spectrometry analysis and K.M. analyzed the data. T.K. prepared the reaction mixture for cell-free protein synthesis system. All authors discussed the results and commented on the paper.

## Competing interests
T.K. is employed by GeneFrontier Corporation. Remaining authors declare no competing interests.
