## [Peer Review File · Communications Biology]

Reviewers' comments:

Reviewer #1 (Remarks to the Author):

Summary

In vitro reconstitution and building of functional ribosomes is a field that has been advancing largely in stepwise fashion. The technique of integrated ribosomal synthesis, assembly, and translation (iSAT) was the first method developed to enable transcription and assembly of functional ribosomes all in the same vessel. However, this method has required the use of crude extracts and ribosomal proteins purified in bulk from native ribosomes. This work presents a slightly different spin on iSAT in that the authors demonstrate the use of individually purified ribosomal proteins added into iSAT system to enable assembly of functional 30S subunits. The primary benefit of individually-purified components of the translation apparatus is the complete control it enables over the environment of ribosomal subunit assembly, and this work logically highlights that advantage by showing the effects of leaving out individual r-proteins on ribosome assembly and reporter protein production. They also look at temporal effects, demonstrating that some 30S r-proteins must be fully synthesized before transcription of the 16S occurs to enable the assembly of 30S particles, while others may be co-transcriptionally produced and still result in functional small subunits.

While citation 17 of this paper (from the same lab) already presented assembly of 30S subunits in the PURE system from individually-purified r-proteins, this work is a little different in that it uses an extract-based system. This would be expected to work, but the data overall seems scientifically sound, and this provides another technique in the toolbox of in vitro translation engineering. I did find some of the data presented a little odd, as described below, but I can recommend the publication of this work assuming a few key issues are addressed and assuming the editors are satisfied with the impact of the work.

Major comments:

- Annotated Genbank files with the DNA sequences of the plasmids used in this study should be provided so that labs which would like to replicate results or build on this work can do so. In particular, the plasmid for 16S rRNA expression is critical. Alternatively, the authors could deposit the plasmid on Addgene, or direct the reader to where the plasmid may be acquired if it is already available.
- The authors write on page 10 line 305: "In contrast to the 50S subunit, which is non-functional when reconstituted with in vitro transcribed rRNA..." It is true that IV transcribed 23S rRNA without modifications does not form functional 50S particles with TP50. However, 23S rRNA transcribed IV in the presence of S150 extract, such that appropriate post-transcriptional modifications occur, does form functional 50S particles, as is the basis for iSAT. It would be useful for the authors to use more precise language to clarify this distinction.
- In Figure 3, synthesis of sfGFP using wild-type ribosomes and wild-type mRNA (green box, blue bar) displays lower fluorescence than any of the orthogonal pairs. This is a curious and unexpected result. Why should wild-type ribosomes have such poor sfGFP expression? Is this just a quirk of this particular experiment and not a general result? This is a very unexpected result that should be justified in the text, as it may call into question the overall reliability of the data in this figure.

Minor comments

- In Figure 1A, it would be useful for understanding the methodology to depict that 50S particles are added fully-assembled into the R-iSAT reaction, and not synthesized alongside the 30S particle. They could be depicted as being added with the "Recombinant rps" in the diagram.
- On page 1, lines 54-56: "Applied studies have also been performed to evolve ribosomes with improved functions or altered chemical properties" It would be appropriate here to cite a new paper from the Jewett group on bioarxiv which has used iSAT in a new strategy for directed evolution of the ribosome - In vitro ribosome synthesis and evolution through ribosome display

Reviewer #2 (Remarks to the Author):

The article by Shimojo M et al. presents a development of the in vitro system based on the PURE cell-free mixture of small ribosome subunit reconstitution and its further testing in one-pot setup. Being largely logical continuation of studies (refs. 7 and particularly 15) authors further improved the reconstitution protocol by coupling transcription of 16S rRNA and its assembly with TP30 that is a pool of proteins stripped from the native small subunit (SSU) of E.coli. However, the significant contribution of this work is in using ribosomal proteins that are translated in the same mixture (by the WT ribosome), and thus newly assembled SSU can be reconstituted with them. To reach this goal, authors sequentially addressed a number of problems as follows.

Initially, the authors explored the possibility of using recombinantly produced ribosomal proteins (R-rps) for SSU reconstitution in vitro. As opposed to the previous study (ref 15 in the article) they used sfGFP, which is to me an adequate control for ribosome activity than polyPhe-synthesis. The activity of reconstituted ribosome with R-rps is less than 10% of the control it was still distinct from the background (Fig1A). Next, authors successfully coupled SSU reconstitution with in vitro production of the 16S rRNA.

Here I have a few minor comments. (1) With regards to data representation, I think it is reasonable to present graphs with a background. However, to a reader, it may be harder to calculate relative ratios of ribosome activity. Given the importance of the background, maybe splitting of the axis would help for calculations. (2) Also, it seems that kinetics of the ribosome assembly is very slow such that even after 4 hours it does not plateau. It would be interesting to see the full extent of the reaction (as a supplementary figure, for instance). (3) Importantly, based on the Fig.1B and Supplementary Fig.3, coupling rRNA transcription with the ribosome reconstitution improves the final sfGFP counts by approximately 1.5-2 fold. Authors did not comment on that at all in the text. This information is essential and may be skipped if not pointed on by authors. (4) Also, given the relatively low yield of ribosome activity, have the authors done any titration of R-rps, or biogenesis factors or they tried only reported concentrations (0.5uM and 0.125uM respectively). It could be a piece of valuable information to readers that can be added to a supplementary. (5) It is unclear also if the wt 16S rRNA sequence was used or the mutant of it reported earlier (ref.7 in the paper)

Then the authors explored the reason for the low yield of reconstitution that potentially was the absence or low level of modification of recombinant ribosomal proteins. Also, ribosome biogenesis factors have been used to improve the yield of SSU assembly.

I have minor suggestions with regards to the text: (1) delete sentence 161 and 162 as it is misleading and data (as authors pointed out in paragraph later) suggests that there is no modification on the tested proteins. (2) General comment: it would be useful and add to the quality of the paper if the sentences in the abstract will be shorter and thus easier to follow.

Having a working system in hands, authors decided to test if (1) skipping certain ribosomal proteins would align with in vivo knockout studies. While most of the data agree with literature, there are examples (S5 and S7) which deletion still ended up with active ribosome though with reduced level. Have authors tested S1 protein here and in later work? It is an interesting example that may improve the yield of translation by promoting initiation step. Also, the presence of S1 may affect ribosome:mRNA orthogonality (that is tested later).

To assess the activity of assembled in vitro SSU it is crucial to differentiate its activity from the SSU that is already present in the mixture (and used for the production of ribosomal proteins). For this authors decided to test if there are SD/anti-SD interaction that can be exploited for it. A reverse-

complement version of the WT SD/anti-SD ('rRNA-comp/mRNA-comp') pair was chosen as the most orthogonal.

(1) While it is hard to argue why comp-version was chosen, it seemed to me that rRNA-9 is more active and thus the ratio of '9' against 'wt' may be more significant. (2) Also, a general and important comment: practically all of data on the bar graphs is normalised to a control translation reaction value. It is, supposedly comes with its own standard deviation. Have the authors propagated SDs or simply divided each replicate to an average value of a control reaction? This has to be explicitly noted in the text. (3) Minor comment: It may be better to use host SSU or any other name authors may come up with but not - 'old', as it sounds confusing.

Next, authors tested if in situ translated ribosomal proteins can be inserted into in situ assembled ribosomes. While most of the tested proteins have resulted in insertion (based on the translation of mRNA-comp) of ribosomal proteins, a few were not functional. Later experiment suggested that is due to the kinetics of ribosome maturation as all of the 'non-functional' proteins are primary binders in the ribosome biogenesis process except bS6. Thus, proteins need to be synthesised before the assembly of the ribosome.

The questions here I have. (1) Have authors measured (by western-blot for example) the level of expression of in situ produced R-rps is in the R-iSAT? This especially applies for problematic proteins S6, S7, S8 and S15. I suppose It would make their arguments with the time-dependent insertion of ribosomal proteins sound if they show that all the tested proteins expressed to equally efficient.

Finally, the authors tested if mutagenesis of the uS12 would lead to the production of streptomycin-resistant or streptomycin-dependent activity of ribosomes. Surprisingly, all of the tested mutants resulted in assembling of streptomycin-resistant ribosomes, and no dependency was observed.

Overall, the work has great potential in further applications. It represents a logical step in building ribosome reconstitution system that is getting incrementally independent from the natively produced ribosomal components whereby initially used modified rRNA and TP30 are ultimately going to be replaced with in vitro transcribed rRNA and in vitro translated ribosomal proteins. This work promises to be a platform for the mutational study of both rRNA and ribosomal proteins; thus both molecules can be mutagenised and coevolved together. I feel positive for recommending it for publication after minor improvement suggested here.

Response to comments of Reviewer #1

We thank the reviewers for considering our work of interest and appreciate constructive suggestions for improving our manuscript. We have made many changes in the revised manuscript. **Data Availability statement** was appended according to the author instructions in *Communications Biology* (P. 14, lines 451-456). We should note that we are sorry that there was a mistake on **Fig. 4a** in the previous submission, where the sequence of mRNA-comp was mistakenly depicted. We revised the figure (P. 21) for correct depiction of the binding between mRNA-comp and rRNA-comp in **Fig. 4a**. Other changes have been made according to the reviewers' suggestions. We noted detailed answers for each concern of the reviewers as below. **Blue sentences** are the original concerns raised by the reviewers and **red words** indicate the revised positions or figures in the main text. Changes in the main text are also noted in **red**.

1-1. Major concern 1 raised by Reviewer #1 is as follows:

Annotated Genbank files with the DNA sequences of the plasmids used in this study should be provided so that labs which would like to replicate results or build on this work can do so. In particular, the plasmid for 16S rRNA expression is critical. Alternatively, the authors could deposit the plasmid on Addgene, or direct the reader to where the plasmid may be acquired if it is already available.

Our answers are as follows:

We thank the reviewer for pointing out the inadequate points in our manuscript. We added a sequence data set including 16S rRNA, sfGFP, and ribosomal proteins for the preparation of DNA templates for R-iSAT in **Supplementary Data 2** with visual annotations using underlines, highlighting, and colored characters. DNA primers for PCR amplification of the DNA templates were also appended in **Supplementary Data 3** for the reproducibility. Accordingly, we added the legends for the **Supplementary Data 2 and 3** (P. 44, lines 917-930) and modified **Online Methods** section (P. 25, lines 686-688 and P. 26, lines 689-704). For the distribution of the plasmids, we are planning to deposit all of the plasmids on Addgene in near future but we are sorry that it is not available at

present. We instead noted that we can share the plasmids on requests in **Online Methods** (P. 26, lines 703-704).

1-2. Major concern 2:

The authors write on page 10 line 305: “In contrast to the 50S subunit, which is non-functional when reconstituted with in vitro transcribed rRNA...” It is true that IV transcribed 23S rRNA without modifications does not form functional 50S particles with TP50. However, 23S rRNA transcribed IV in the presence of S150 extract, such that appropriate post-transcriptional modifications occur, does form functional 50S particles, as is the basis for iSAT. It would be useful for the authors to use more precise language to clarify this distinction.

Our answers:

We are sorry for the immature discussion. We modified the corresponding sentences to clarify the current status of the methodology for in vitro 50S subunit reconstitution (P. 11, lines 353-357).

1-3. Major concern 3:

In Figure 3, synthesis of sfGFP using wild-type ribosomes and wild-type mRNA (green box, blue bar) displays lower fluorescence than any of the orthogonal pairs. This is a curious and unexpected result. Why should wild-type ribosomes have such poor sfGFP expression? Is this just a quirk of this particular experiment and not a general result? This is a very unexpected result that should be justified in the text, as it may call into question the overall reliability of the data in this figure.

Our answers:

We thank the reviewer for pointing out this important observation. First of all, the observation was not just a quirk of the experiment but a reproducible result. We additionally appended **Supplementary Fig 8** (P. 38), showing the time-course of sfGFP expression where the efficiency of orthogonal pairs can be seen in an intuitive way. The data indicated that those orthogonal pairs may have facilitated ribosome assembly or

translation processes. We have two explanations for this result. The first one is that the three pairs from the literature of Prof. Chin J.W. were selected through *in vivo* evolutionary experiments where a plenty of wild-type ribosomes co-existed in cells. We consider that this scheme might have resulted in the selection of sequences that are favorable for the assembly or translation efficiency. The second one is from the view of the ribosome evolution. Because the mutation in the anti-SD sequence may affect all gene expression in cells, there is a possibility that this region may have been left out during the evolution, leaving room for improving ribosome functions through the engineering of the anti-SD region. Accordingly, we mentioned this observation in the main text (P. 8, lines 243-257).

1-4. Minor concern 1:

In Figure 1A, it would be useful for understanding the methodology to depict that 50S particles are added fully-assembled into the R-iSAT reaction, and not synthesized alongside the 30S particle. They could be depicted as being added with the “Recombinant rps” in the diagram.

Our answers:

Fig. 1a (P. 19) and **Fig. 4a** (P. 23) were modified according to the suggestion.

1-5. Minor concern 2:

On page 1, lines 54-56: “Applied studies have also been performed to evolve ribosomes with improved functions or altered chemical properties” It would be appropriate here to cite a new paper from the Jewett group on bioarxiv which has used iSAT in a new strategy for directed evolution of the ribosome - In vitro ribosome synthesis and evolution through ribosome display.

Our answers:

We thank for this information and the corresponding manuscript, Hammerling *et al.*, (P. 15, lines 502-504) was cited in three places in the main text (P. 3, line 55; P. 4, line 94; P. 14, line 442).

Response to comments of Reviewer #2

We thank the reviewers for considering our work of interest and appreciate constructive suggestions for improving our manuscript. We have made many changes in the revised manuscript. **Data Availability statement** was appended according to the author instructions in *Communications Biology* (P. 14, lines 451-456). We should note that we are sorry that there was a mistake on **Fig. 4a** in the previous submission, where the sequence of mRNA-comp was mistakenly depicted. We revised the figure (P. 21) for correct depiction of the binding between mRNA-comp and rRNA-comp in **Fig. 4a**. Other changes have been made according to the reviewers' suggestions. We noted detailed answers for each concern of the reviewers as below. **Blue sentences** are the original concerns raised by the reviewers and **red words** indicate the revised positions or figures in the main text. Changes in the main text are also noted in **red**.

2-1-1. Minor concern 1 raised by Reviewer #2 is as follows:

(1) With regards to data representation, I think it is reasonable to present graphs with a background. However, to a reader, it may be harder to calculate relative ratios of ribosome activity. Given the importance of the background, maybe splitting of the axis would help for calculations.

Our answers are as follows:

Fig. 1b and 1c were modified to show only the fluorescence intensities originated from synthesized sfGFP for the readability. We instead made it clear that we subtracted the background intensity in the legend for **Fig. 1** (P. 19). The newly appended figures during this revision, including **Supplementary Fig. 1** and **Supplementary Fig. 8**, are shown in a similar way with **Fig. 1**. We also noted that the background intensity was subtracted in the legend for those figures (P. 30 and P. 38).

2-1-2. Minor concern 2:

(2) Also, it seems that kinetics of the ribosome assembly is very slow such that even after 4 hours it does not plateau. It would be interesting to see the full extent of the

reaction (as a supplementary figure, for instance).

Our answers:

We additionally performed R-iSAT experiment with long-term incubation for 12 h, which exhibited almost plateau level of sfGFP synthesis. The results are shown as **Supplementary Fig. 1** (P. 30) and mentioned in the main text (P. 6, lines 154-155).

2-1-3. Minor concern 3:

(3) Importantly, based on the Fig.1B and Supplementary Fig.3, coupling rRNA transcription with the ribosome reconstitution improves the final sfGFP counts by approximately 1.5-2 fold. Authors did not comment on that at all in the text. This information is essential and may be skipped if not pointed on by authors.

Our answers:

We thank the reviewer for pointing out this important aspect. We consider that the coupling of 16S rRNA transcription in the presence of ribosomal proteins are beneficial for correct folding of 16S rRNA, which is also suggested by the recent single molecule imaging studies (Duss, O. *et al.*, *Cell*, 2019, **179**, 1357; Rodgers, M.L. *et al.*, *Cell*, 2019, **179**, 1370). We mentioned this aspect with these references (P. 16, lines 526-530) in the main text (P. 5, lines 149-151 and P. 6, lines 152-154; P. 6, lines 177-178).

2-1-4. Minor concern 4:

(4) Also, given the relatively low yield of ribosome activity, have the authors done any titration of R-rps, or biogenesis factors or they tried only reported concentrations (0.5uM and 0.125uM respectively). It could be a piece of valuable information to readers that can be added to a supplementary.

Our answers:

We've already seen the concentration dependencies of these factors on sfGFP synthesis in R-iSAT. However, the experiments were preliminary to be shown in the manuscript with a few replicates with different series of concentrations. Therefore, we additionally

performed experiments to show these points. The results shown in **Supplementary Fig. 2** indicates that the concentrations used in this manuscript were within the range that almost maximize the sfGFP synthesis in R-iSAT. Thus, we appended **Supplementary Fig. 2** (P. 31) and mentioned them in the main text (P. 6, lines 155-157; P. 6, lines 172-173).

2-1-5. Minor concern 5:

(5) It is unclear also if the wt 16S rRNA sequence was used or the mutant of it reported earlier (ref.7 in the paper)

Our answers:

We are sorry for this unclear point. We made it clear in the main text that we used a native sequence derived from *rrnB* operon in *E. coli* genome (P. 5, line 140). We also appended **Supplementary Data 2** which includes all of the sequences used for R-iSAT experiments, according to the suggestion by Reviewer #1 (Major concern 1-1).

2-2-1. Minor concern 6:

(1) delete sentence 161 and 162 as it is misleading and data (as authors pointed out in paragraph later) suggests that there is no modification on the tested proteins.

Our answers:

We agree with the suggestion and edited out the corresponding sentences (P. 6, lines 168-169).

2-2-2. Minor concern 7:

(2) General comment: it would be useful and add to the quality of the paper if the sentences in the abstract will be shorter and thus easier to follow.

Our answers:

We thank for the suggestion and accordingly edited the abstract section for simple description of our present study (P. 2).

2-3-1. Major concern 8:

Have authors tested S1 protein here and in later work? It is an interesting example that may improve the yield of translation by promoting initiation step. Also, the presence of S1 may affect ribosome:mRNA orthogonality (that is tested later).

Our answers:

We agree with the reviewer's suggestion and performed additional experiments for clarifying the effect of bS1 removal and expression. The results were little bit complicated. Firstly, we did not see strong dependency of bS1 on sfGFP expression from canonical SD/anti-SD pair (**Supplementary Fig. 6a; P. 36**), whereas the use of orthogonal SD/anti-SD pair strongly depended on the presence of bS1 (**Supplementary Fig. 6b; P. 36**). In addition, we have previously prepared bS1-free ribosomes and examined protein expression from mRNA with 5' leader sequence, where bS1-free ribosomes were not functional in the absence of bS1 (Qi, H. *et al.*, *J. Mol. Biol.*, 2007, **368**, 845). The used 5' leader sequence was very different from that used in this experiment (5'-GGGAAUUUUUUUAUCGGGAAAUCUCAUG... in JMB experiment; 5'-GGGAGACCACAACGGUUUCCCUCUAGAAAUAUUUUUGUUUAACUUUAAGAAGGA GAUAUACCAAUG... in this manuscript). From these results, we consider that the bS1 dependency may depend on the selected leader sequence including SD sequence, although the details should be clarified with additional experiments. These points were added in the main text (**P. 7, lines 203-207; P. 10, lines 310-319**) with a newly cited reference (**P. 16, lines 552-553**).

Secondly, the effect of bS1 expression was comparatively lower than other proteins (**Supplementary Fig. 6b**). We considered that this is due to the competition among RNAs to be transcribed (rRNA, sfGFP mRNA, and bS1 mRNA) and therefore, performed an experiment with sufficient amount of bS1, resulting in decrease of sfGFP synthesis when bS1 template is added (**Supplementary Fig. 6b**). Because the size of bS1 is much larger than other proteins, degree of this competition might become higher and thus, the effect of bS1 expression might have been constrained. Nevertheless, we observed slight stimulation of sfGFP synthesis when the bS1 expression was performed. These points

were added in the main text (P. 10, lines 301-309).

2-4-1. Major concern 9:

(1) While it is hard to argue why comp-version was chosen, it seemed to me that rRNA-9 is more active and thus the ratio of '9' against 'wt' may be more significant.

Our answers:

We are sorry for our complicated experiments. The expression of sfGFP should be under the control of orthogonal SD sequence and the expression with the wild-type ribosomes is unwelcome as depicted in **Fig. 4a**. By contrast, we don't care about the expression of ribosomal proteins from the newly-assembled ribosomes, *i.e.*, the expression with orthogonal rRNA/wild-type mRNA pair, at this stage, because it may not affect the detection with fluorescence. Therefore, the most important ratio for our purpose is the ratio of sfGFP intensity with orthogonal rRNA/orthogonal mRNA pairs against the intensity with wild-type rRNA/orthogonal mRNA pairs. The corresponding ratios are 31.5, 16.0, 27.7, and 56.2 for '2', '8', '9', and 'comp' rRNA, respectively, which suggests that the use of 'comp' reasonable for subsequent experiments shown in **Fig. 4** and **5**. We appended a graph showing these ratios in **Supplementary Fig. 7** (P. 37) and mentioned in the main text (P. 8, lines 228-229; P. 8, lines 238-239). We also note that the important observation that the use of orthogonal rRNA is more efficient for sfGFP synthesis is mentioned in P. 8, lines 243-257, according to the suggestion of Reviewer #1 (Major concern 1-3).

2-4-2. Major concern 10:

(2) Also, a general and important comment: practically all of data on the bar graphs is normalised to a control translation reaction value. It is, supposedly comes with its own standard deviation. Have the authors propagated SDs or simply divided each replicate to an average value of a control reaction? This has to be explicitly noted in the text.

Our answers:

We are sorry for the complicated description. We chose the normalization by dividing by

the average value of the control reaction. The legend for **Fig. 2** (P. 20), **Fig. 3** (P. 21-22), **Fig. 4** (P. 23), **Fig. 5** (P. 24) and **Supplementary Fig. 2** (P. 31), **Supplementary Fig. 4** (P. 34), **Supplementary Fig. 5** (P. 35), **Supplementary Fig. 6** (P. 36), **Supplementary Fig. 11** (P. 41), **Supplementary Fig. 12** (P. 42) were revised accordingly.

2-4-3. Minor concern 11:

(3) Minor comment: It may be better to use host SSU or any other name authors may come up with but not - 'old', as it sounds confusing.

Our answers:

We thank the reviewer for this helpful suggestion. We agree that the use of “host” is better to avoid any confusion and accordingly revised corresponding places (P. 4, line 96; P. 7, line 210; P. 7, line 213; P. 8, line 241 and the legend for **Fig. 3** (P. 21)).

2-5-1. Major concern 12:

The questions here I have. (1) Have authors measured (by western-blot for example) the level of expression of in situ produced R-rps in the R-iSAT? This especially applies for problematic proteins S6, S7, S8 and S15. I suppose It would make their arguments with the time-dependent insertion of ribosomal proteins sound if they show that all the tested proteins expressed to equally efficient.

Our answers:

We have already confirmed the expression of each ribosomal protein in the PURE system. The data showed successful expression of all proteins with yields in the same orders of magnitude. In this experiment, we also examined the aggregate formation of the synthesized proteins and confirmed that all proteins were recovered in the soluble fractions. Though this is not exactly the same condition as R-iSAT where rRNA co-transcription occurs, we agree with the Reviewer’s comment that the results are informative to the readers. Thus, the SDS-PAGE results of cell-free expressed ribosomal proteins are appended in **Supplementary Fig. 9** (P. 39) and mentioned in the main text (P. 9, lines 273-276). Methods for this analysis was appended in **Online Methods** section

(P. 28, lines 768-776).

REVIEWERS' COMMENTS:

Reviewer #1 (Remarks to the Author):

The authors have mostly addressed my concerns with their manuscript. In my opinion, I find the explanation for the data in Figure 3 to be somewhat implausible, but the relative activity of wild-type versus the orthogonal ribosomes in their system is not a key result of the paper. If the editors are satisfied with the explanation, I can recommend publication.

Reviewer #2 (Remarks to the Author):

I have looked into the rebuttal letter as well as revised manuscript. I am satisfied with their reply and I have no further questions or concerns.